# Relying on Incomplete Information Can Lead to the Wrong Conclusions. Comment on van Hassel, F.; Bovenkerk, B. How Should We Help Wild Animals Cope with Climate Change? The Case of the Iberian Lynx. *Animals* 2023, *13*, 453

**DOI:** 10.3390/ani13203245

**Published:** 2023-10-18

**Authors:** Juan F. Beltrán, Eduardo J. Rodríguez-Rodríguez

**Affiliations:** Departamento de Zoología, Universidad de Sevilla, 41012 Sevilla, Spain; edurodrodbio@gmail.com

In their recent paper, Van Hassel and Bovenkerk [1] raised the question of how we should aid wild animals in coping with climate change and they used the Iberian lynx (*Lynx pardinus*) as a case study. In our opinion, this might have been an interesting effort if it were approached with rigor and updated information. However, we found their assessment of the state of Iberian lynx was seriously flawed because the authors relied on limited information on the current status of lynx. This species is likely one of most investigated carnivores, with studies conducted on its ecology and management [2,3,4,5], taxonomy [6], morphology [7,8], physiology, and diseases ([9,10] for a review). These investigations underpinned efforts to address the threat of extinction [10,11]. As a result, the Iberian lynx is probably a good example of how recovery efforts can reverse the rapid decline of an endangered species [12,13].

Van Hassel and Bovenkerk [1] specifically overlooked much of the work conducted in last 10 years, a period that has been critical to the survival of the Iberian lynx. In 2015, the IUCN conservation status of this species changed from Critical Endangered (CR) to Endangered (E) [14], and annual population growth was 18% from 2013 to 2022 (Figure 1). In 2020, 414 cubs were born to 239 adult females, an increase of 30% from the previous year [15]. The most recent information (May 2023) confirmed this steady annual increase with 326 reproductive females, 563 cubs, for a total lynx population of 1668 individuals (Figure 1). Similar changes have occurred in captive populations of lynx. Accordingly, the IUCN conservation status may be downgraded to Vulnerable (V).

Not including recent information has obvious implications to the approach used and subsequent recommendations from their evaluation. For example, referring to models on the effects of climatic change [17] when the total number of Iberian lynxes was 150–200 rather than the current population of ~1700 is clearly problematic. Moreover, references on the effect of climate change on the lynx’s main prey, the European wild rabbit (*Oryctolagus cuniculus*) [18], are also missing. Many of the conservation measures proposed by Van Hassel and Bovenkerk [1] are not original or new (e.g., providing supplementary food [19], assisting migration/colonization, and bringing animals into captive breeding programs [10,11]). Most of them were considered in the five successive LIFE projects funded in part by the European Union (EU) (Figure 2).

The first LIFE project (1994–1998) [20] included a series of actions to conserve and restore the lynx’s habitat, rabbit management, and surveillance and monitoring of areas occupied by lynx to increase knowledge of their threats and causes of mortality. The second LIFE project (2001–2006) [15] was oriented towards population recovery and stabilization. The goal of the third LIFE project (2006–2011) [21] was the reintroduction of the Iberian lynx to some select areas of Andalucía (southern Spain), the region holding two of the more important Iberian lynx populations at that time. The results of these projects included a substantial decrease in extinction risk, coupled with an increase both in Iberian lynx numbers and populations. The fourth LIFE project [21] had a more ambitious goal—restoring the historical distribution of lynx throughout Spain. This was approached by continuing the efforts to increase lynx numbers and populations and further aided in diminishing threats of extinction. The current LIFE project [15] is directed toward reducing the fragmentation among existing populations. As a direct result of these efforts, the present-day status of the Iberian lynx clearly contrasts with that observed two decades ago. The worldwide population has increased 10-fold during last decade, and 20-fold since 2002.

In conclusion, we believe that complete knowledge that is the product of well-designed research projects is essential to guide recovery programs for endangered species [22,23,24]. The successful implementation of conservation measures also requires the fundamental support of society [25,26]. To achieve this, there must be a deliberate transfer of information to the public. We agree with Van Hassel and Bovenkerk [1] that there many species are threatened by climate change. We would extend these threats under a wider perspective of global change, including habitat loss, emergent diseases, and invasive species [27]. Iberian lynx have benefitted from substantial research, conservation actions, and public support. An extraordinary amount of resources have been allocated to bring it from near extinction. We remain optimistic for the recovery of the Iberian lynx.

## Figures and Tables

**Figure 1 animals-13-03245-f001:**
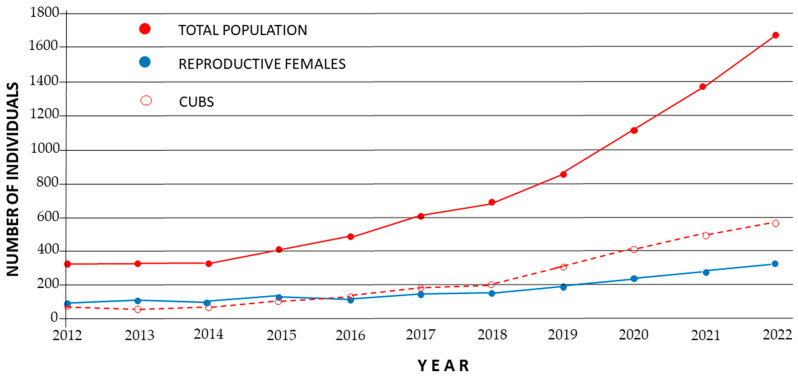
Evolution of Iberian lynx numbers in the wild since 2012. Data from [15,16]. Population mean lambda value = 1.18 (years 2013–2022 mean); adult reproductive female lambda value: 1.15 (2013–2022 mean).

**Figure 2 animals-13-03245-f002:**
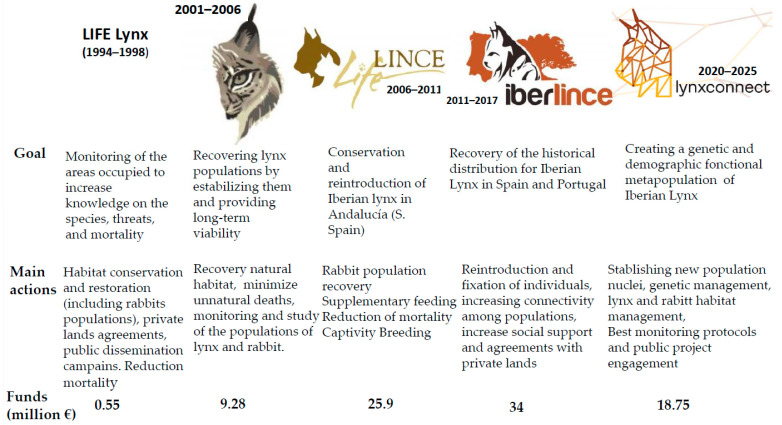
A brief summary of the five LIFE projects carried out on the Iberian lynx from 1994 to the present.

## Data Availability

All data and information are available to the public.

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
