# Peer review of "Relying on Incomplete Information Can Lead to the Wrong Conclusions. Comment on van Hassel, F.; Bovenkerk, B. How Should We Help Wild Animals Cope with Climate Change? The Case of the Iberian Lynx. Animals 2023, 13, 453"

_animals, 2023, doi:10.3390/ani13203245_

Round 1
Reviewer 1 Report
This manuscript represents a rebuttal to the Comment by Van Hassel and Bovenkerk (2023) entitled "How Should We Help Wild Animals Cope with Climate Change? The Case of the Iberian Lynx". The authors strongly feel that this article does not fully represent the facts regarding the recovery actions that have been implemented for the Iberian Lynx and their success and present a convincing body of evidence. I understand that it can be frustrating when third parties publish commentaries on case-studies without a full understanding on the context behind them. I feel that the authors of this manuscript are well-placed to argue that the original Comment does not adequately represent the current status of the Iberian Lynx and, as such, I am happy to recommend publication.
Furthermore, having read Van Hassel and Bovenkerk's Comment, I believe they have acted in good faith and have not deliberately misrepresented the conservation efforts of the Iberian Lynx. I don't necessarily think the Iberian Lynx is necessarily the best species to present a case-study on the potential value of assisted colonisation/migration (which is essentially the crux of Van Hassel and Bovenkerk's rather verbose commentary), but that is not what is being examined in this manuscript. As such, I feel some of the language used is unnecessarily inflammatory (e.g. Lines 27-29 and 88-92) and I would like to see it toned down to be less defensive. While I recognise the authors' frustration, I don't feel that Hassel and Bovenkerk's is 'seriously flawed'. I would also like the authors to explicitly address whether they consider that assisted migration is potentially a valid management action for the conservation of Iberian Lynx.
I identified the following additional points that need addressing prior to publication:
Line 9: 'field' is misspelled
Line 11: "[REF 1]" not sure what this refers to - suggest deleting
Line 17: 'lynx' is missing after 'Iberian'
Line 29: 'lynx' is missing after 'Iberian'
Line 41: IUCN is misspelled
Line 47: A change from Endangered to Vulnerable is considered a 'downgrade' not an 'upgrade' by IUCN
Line 53-54: citations appear to be missing
Line 62: "-cited by the authors-" not sure why this is here - suggest deleting
Line 97: ) missing after [25]
Author Response
We have addressed all the minor suggestions, and activated Word changes in the ms to show them. Point-by-point replies are available in the attached PDF file.
Thank you for your time and suggetions.
Best,
JF Beltrán and E. Rodriguez

Reviewer 2 Report

Minor editing is needed
Author Response
We have addressed all the suggestions, including new references and double-checking all references. Point-by-point replies are available in the attached PDF file. We have activated the Word track changes to show them.
Thanks for your time and suggestions.
Best,
JF Beltrán and E. Rodriguez
